# Synthesis and Characterization of Polyaniline/Carbon Nanodots: Electrochemical Sensing of Alcohols for Freshness Monitoring for Application as Packaging Materials

**DOI:** 10.3390/nano15080593

**Published:** 2025-04-12

**Authors:** Shaila Jackson, Mary Taylor, Rajeev Kumar, Amit Kumar Shringi, TinChung Leung, Ufana Riaz

**Affiliations:** 1Department of Chemistry and Biochemistry, North Carolina Central University, Durham, NC 27707, USAmtaylo95@eagles.nccu.edu (M.T.);; 2Biomedical/Biotechnology Research Institute, North Carolina Central University, Durham, NC 27707, USA; 3Department of Biological & Biomedical Sciences, North Carolina Central University, Durham, NC 27707, USA

**Keywords:** polyaniline, carbon nanodots, X-ray photoelectron spectroscopy, electrochemical sensing

## Abstract

The increasing demand for innovative, sustainable, and functional food packaging solutions has led to the exploration of polyaniline (PANI) and carbon nanodots (CNDs) as promising materials for smart packaging. This study investigates the potential of PANI/CND composites for electrochemical sensing of alcohols, a key indicator of spoilage in food products. By leveraging the unique properties of PANI, a conducting polymer, and the fluorescence and electrochemical activity of CNDs, we propose a dual-function smart packaging material capable of real-time monitoring of alcohol levels, which are often released during the fermentation or degradation of food. The integration of PANI with CNDs enhances the material’s sensitivity and stability, offering a cost-effective, environmentally friendly, and responsive solution for freshness and spoilage detection. The electrochemical sensing mechanism allows for rapid, non-destructive testing, providing consumers and food producers with an efficient tool to assess product quality and safety. This work paves the way for the development of intelligent packaging systems that can not only detect spoilage but also actively contribute to food preservation, reducing waste and improving food safety.

## 1. Introduction

As the demand for clean, sustainable, and renewable energy continues to rise, there is a parallel need for advanced smart food packaging that can effectively support the preservation, safety, and quality of food products [1,2]. Smart packaging technologies, which incorporate sensors, indicators, and other advanced materials, are crucial for enhancing food storage systems, ensuring longer shelf life, improving food safety, and maintaining nutritional value. Smart food packaging systems are increasingly favored for their ability to monitor and manage the conditions of food products in real time, offering advantages such as enhanced freshness, improved shelf-life, and reduced food waste [3,4]. Intelligent packaging is designed to monitor the condition of the food product throughout its storage and transportation life to detect changes in factors such as temperature, humidity, gases, or pH levels, providing valuable information about the product’s quality and safety [5,6,7,8].

When food spoils, various alcohols such as ethanol, methanol, butanol, and isopropanol are produced due to microbial metabolism, fermentation, and enzymatic breakdown of organic compounds. Ethanol is the most common alcohol formed, especially in carbohydrate-rich foods due to yeast fermentation, while methanol can be found in spoiled fruits and vegetables due to pectin degradation. Butanol and isopropanol are byproducts of bacterial metabolism in spoiled dairy, meat, and anaerobic environments. Food spoilage occurs due to microbial growth, chemical reactions (such as oxidation), and environmental factors like temperature, humidity, and oxygen exposure, leading to changes in texture, odor, color, and taste. Various technologies exist to detect spoilage, including gas sensors that monitor volatile organic compounds (VOCs) like ethanol, ammonia, and hydrogen sulfide, which indicate bacterial activity. pH sensors detect acidity changes caused by microbial fermentation, while electrochemical biosensors identify specific metabolites such as biogenic amines in spoiled meat and fish. Optical sensors, including colorimetric indicators and fluorescence-based systems, react to spoilage markers by changing color, often integrated into smart packaging. Electronic noses (E-noses) use sensor arrays to analyze spoilage-related odors by detecting aldehydes, ketones, and sulfur compounds. These technologies, combined with advancements in smart packaging, are improving real-time food safety monitoring and extending shelf life by providing early warnings of spoilage.

Conducting polymers have been widely popular for their electroactive properties and sensing characteristics, and hold immense potential for application in corrosion protective coatings [9,10,11,12] photocatalysis [13,14]. Polyaniline’s (PANI) conductive nature makes it ideal for packaging applications where electrochemical activity is critical. PANI in its emeraldine salt form, is known for its excellent stability in both air and moisture, making it a reliable choice for packaging materials exposed to diverse climates [15,16,17]. PANI can be synthesized into flexible films or coatings, which makes it ideal for applications requiring lightweight, durable, and bendable packaging materials [18,19,20]. This flexibility is particularly valuable in wearable electronics, portable energy storage devices, and other applications where compact and lightweight designs are essential. The mechanical strength of polyaniline also helps protect the internal components of the energy storage device from physical stress.

To enhance the mechanical and electrochemical properties of PANI, the present work reports the synthesis of carbon nanodots (CND) and dispersed PANI composites via ultrasonication. The composites were characterized by their spectral, morphological, and electrochemical properties. The sensing studies were carried out for pure PANI and CND/PANI composites against ethanol to explore the improvement in the sensing upon the loading of CNDs, which can hold potential application as smart packaging materials.

## 2. Materials and Methods

Aniline monomer (Sigma Aldrich, St. Louis, MO, USA), ferric chloride (Sigma Aldrich, St. Louis, MO, USA), urea (Fisher scientific, Raleigh, NC, USA), and citric acid (Fisher scientific, Raleigh, NC, USA) were purchased and used without further purification.

### 2.1. Synthesis of CNDs

Carbon nanodots (CNDs) were synthesized using a microwave-assisted method with citric acid and urea serving as the carbon and nitrogen sources, respectively. Equimolar concentrations of citric acid and urea were prepared by dissolving 3 g (1 mole each) of both in 10 mL of deionized (D.I.) water to create a transparent solution. This solution was transferred to a 100 mL conical flask and exposed to microwave radiation at 700 W for approximately 15 min. The microwave treatment produced a brown aggregated cluster, whose volume increased over time. The cluster was crushed and dried in a vacuum oven at 80 °C for 1 h to remove moisture and volatile gases, such as ammonia, generated during the synthesis. The resulting brown powder was cooled to room temperature, dispersed in water, and formed a black murky suspension. This liquid was then purified through two successive steps of centrifugation and filtration using a 10 kDa filter, yielding a pale-yellow aqueous solution of nitrogen-functionalized carbon nanodots (N-CDs).

### 2.2. Synthesis of PANI

Sulfate-doped polyaniline (PANI) was synthesized using a chemical oxidative polymerization method in an acidic medium. The process began by dissolving aniline monomer in a solution of sulfuric acid, which served both as the acid medium and the source of sulfate dopant ions. The typical concentration of sulfuric acid was adjusted to ensure sufficient protonation of the aniline monomers and effective doping during polymerization. Ammonium persulfate (APS) was used as the oxidizing agent to initiate the polymerization reaction. A pre-cooled solution of APS, prepared in deionized water, was added dropwise to the aniline-sulfuric acid solution under constant stirring at a temperature maintained between 0 and 5 °C to control the reaction rate and avoid side reactions. The polymerization process was allowed to proceed for several hours, typically 4–6 h, during which the reaction mixture turned dark green, indicating the formation of the conductive emeraldine salt form of polyaniline. The resulting sulfate-doped PANI precipitate was collected by filtration or centrifugation and washed thoroughly with deionized water and ethanol to remove unreacted monomers, oxidants, and byproducts. Finally, the product was dried in a vacuum oven at 50–60 °C to obtain the sulfate-doped PANI as a dry powder.

### 2.3. Synthesis of CND/PANI Nanocomposites

For the composite preparation, a calculated amount of PANI powder and CNDs was mixed in a specific weight ratio (1:1 and 1:2). The composites were designated as 1-CND/PANI and 2-CND/PANI based on the loading of the CNDs. The mixture was placed into a ball-mill or a mechanochemical mixer and subjected to high-energy mechanical grinding for 1–2 h. The grinding process facilitates the physical integration of CNDs into the PANI matrix, ensuring uniform dispersion of the nanodots within the polymer structure. During the mechanochemical mixing, strong shear forces and collisions enhance the interaction between the CNDs and PANI chains, potentially creating non-covalent interactions such as π-π stacking or hydrogen bonding. This integration enhances the composite’s electrical conductivity, mechanical strength, and surface properties.

### 2.4. Characterization

#### 2.4.1. Spectral Studies

Fourier transform infrared spectroscopy (FTIR) spectra were collected in a Perkin Elmer Spectrum One spectrometer in transmission mode at a resolution of 4 cm^−1^. UV-Visible Spectroscopy was collected on a Shimadzu UV-Visible spectrometer (Kyoto, Japan). The fluorescence emission spectra were recorded in the wavelength range of 280 nm–550 nm on a fluorescence spectrophotometer model Fluorolog^®^3, Horiba Scientific, Irvine, CA, USA. Rayleigh Light scattering (RLS) measurements were performed via simultaneously scanning the excitation and emission monochromators of the spectrofluorometer from 50 nm to 400 nm with ∆λ = 0 nm and a slit width of 1.5 nm. X-ray photoelectron spectroscopy (XPS) spectra were collected using a Physical Electronics Versa Probe II instrument (Physical Electronics GmbH, Feldkirchen, Germany) equipped with a monochromatic Al Kα X-ray source (hν = 1486.7 eV) and a concentric hemispherical analyzer. XPS spectra were recorded and surface elemental stoichiometries were determined from the peak area ratios, which were corrected using the experimentally determined sensitivity factors. The carbon 1s (C 1s), oxygen 1s (O 1s), and nitrogen 1s (N 1s) spectra were fitted with Gaussian component peaks, each having the same full width at half maximum (FWHM). The positions and intensities of the component peaks were optimized to achieve the best fit to the experimental data.

#### 2.4.2. Morphological Analysis

Powder X-ray Diffraction (PXRD) data were collected on a Malvern Panalytical Empyrean diffractometer (Malvern, UK). The instrument was equipped with a Cu-Kα, operated at a voltage of 45 keV and a power of 40 kW. Patterns were obtained utilizing reflection mode and a PIXcel 3D detector. An FEI XL30 SEM-FEG Scanning Electron Microscope (FEI, Hillsboro, OR, USA) was used to examine the morphology and elemental mapping via energy dispersive X-ray (EDX) analysis.

#### 2.4.3. Electrochemical Measurements

All electrochemical measurements were carried out in three electrode configurations, using WaveDriver 200 bipotentiostat (2741 Campus Walk Avenue, Building 100, Durham, NC 27705, USA, Pine Research). A total of 5 mg of PANI and CND/PANI samples were dispersed in 900 µL of IPA and 100 µL of DI water. A total of 100 µL of Nafion binder (5 wt.% in IPA) was added. A total of 5 mg of PANI-PSS sample was dispersed in 900 µL of IPA and 100 µL of CND solution. A total of 100 µL of Nafion binder (5 wt.% in IPA) was added. A total of 2.5 µL of the above dispersions were drop-cast on a glassy carbon (working) electrode. The reference electrode was Ag/AgCl and the counter was Pt wire. The electrolyte was 0.5 M H_2_SO_4_. Ar gas was purged prior to electrochemical measurements. Cyclic voltammograms were recorded at a scan rate of 5 mV/s in the potential window of 0.5 to −0.5 V vs. Ag/AgCl. Differential pulse voltammetry (DPV) measurements were recorded in the potential window of −0.1 to −0.6 V vs. Ag/AgCl. The pulse parameters were set as height-50 mV, width-0.01 s, period-0.1 s, and increment-10 mV. The DPV curves were recorded without cathodic inversion. Thus, all current responses appear in the negative region, unlike conventional positive responses in most reports. A few µL of ethanol was periodically added and mixed well in the electrolyte. The %*v*/*v* of ethanol in the electrolyte was evaluated. The electrochemical measurements were recorded to evaluate the sensing behavior.

## 3. Results

### 3.1. FTIR Analysis

The FTIR spectrum of CND, Figure 1, revealed peaks at 3310 cm^−1^, 1650 cm^−1^, and 1031 cm^−1^ corresponding to the N of urea, C=O of urea, and OH bending due to citric acid used as a precursor for the synthesis of CND [21]. The FTIR spectrum of pure PANI revealed an NH stretching vibration peak at 3230 cm^−1^, while the peaks for benzenoid and quinonoid vibrations were noticed at 1550 cm^−1^, 1490 cm^−1^, 1466 cm^−1^, and 1457 cm^−1^. The imine stretching peak was found at 1620 cm^−1^ while the CN stretching vibration peak was noticed at 1257 cm^−1^. Pekas due to the benzene ring appeared at 950 cm^−1^, 857 cm^−1^, and 750 cm^−1^. The FTIR spectrum of 1-CND/PANI revealed a broad hump at 3305 cm^−1^ due to the presence of NH of PANI, as well as NH of urea present in CND while the C=O stretching peak, due to CND, appears at 1647 cm^−1^. The peaks at 1559 cm^−1^, 1455 cm^−1^, 1437 cm^−1^, and 1401 cm^−1^ corresponded to the presence of quinonoid and benzenoid rings in PANI [16] while the OH bending peak of CND was observed at 1036 cm^−1^. The aromatic stretching vibrations were seen at 847 cm^−1^ and 673 cm^−1^ and the presence of PANI in CND was confirmed. The FTIR spectrum of 2-CND/PANI also revealed the same peaks since the loading of CND was slightly higher and showed minor shifts in the peaks associated with PANI.

### 3.2. UV-Visible Analysis

UV-Vis spectra of CND and CND/PANI composites are shown in Figure 2. Pure CND absorbed mostly in the UV region due to its graphitic aromatic structure. The absorption peak for pure CND was noticed between 200 and 300 nm, indicating π→π* transition [22]. The 1-CND/PANI showed increased absorbance in the visible region as compared to pure CND. Absorption peaks were noted around 400–600 nm due to π→π* transition in the quinone ring of PANI. The 2-CND/PANI showed higher absorbance than 1-CND/PANI in the visible region, suggesting that 2-CND/PANI exhibited synergistic interaction between CND and PANI. The introduction of PANI to the composite extended the absorption into the visible range (400–700 nm).

### 3.3. XRD Analysis

The XRD profile of CND, shown in Figure 3, showed a pronounced peak at 26.3° corresponding to the 002 plane, and matches well with the literature values [23]. The XRD profile of 1-CND/PANI shows peaks at 2θ = 14.03°, 18.33°, 21.05°, and 28.05°, while 2-CND/PANI showed peaks at 2θ = 14.05°, 18.39°, 21.30°, and 28.03° corresponding to 001, 011, and 020 200 planes of PANI and 002 planes of CND. The 002 plane shows a shift in PANI/CNDs, reflecting a decrease in the lattice parameter of CND due to the insertion of PANI. The presence of well-formed peaks reveals that PANI is crystalline; the peak corresponding to CND diminished in the presence of PANI, indicating that CND was encapsulated by PANI.

### 3.4. Fluorescence Studies

The pristine CND, shown in Figure 4, showed peaks at 460 nm and 600 nm [24]. The 1-CND/PANI showed an increase in the peak intensity of 460 nm compared to PANI, while the peak at 600 nm in pure CND showed a shift towards 560 nm. This confirmed that the presence of PANI enhanced the fluorescence intensity but shifted the emission towards a lower wavelength. The 2-CND/PANI revealed the higher intensity as compared to 1-CND/PANI due to higher loading of CND, and higher synergistic interaction between CND and PANI due to higher loading of the former.

### 3.5. XPS Studies

The survey spectra of CND and CND/PANI, shown in Figure 5a–k, helped in identifying the presence of C 1s and O 1s in pristine CND and the presence of C 1s and O 1s as well as N 1s due to PANI in CND/PANI composites. The binding energy (eV), corresponing to its specific peak, are label based on the elements oxygen (O), nitrogen (N), and carbon. The O 1s peaks labeled around ~530 eV indicated oxygen-containing groups on the surface. The C 1s peaks were seen around ~285 eV, indicating that the material contained a significant amount of carbon. On the spectrum, the N 1s peaks were found in the PANI composites around ~400 eV. The XPS of the C1s spectrum of CND, shown in Figure 5b, showed peaks at 284.82 eV, 284.77 eV, 285.73 eV, and 285.93 eV due to the presence of C–C and C–OH bonds in CND. The peaks at 288.12 eV, 288.42 eV, 289.47 eV, and 289.72 eV were due to the presence of O–C=O and π-π* bonding [25,26,27,28,29]. The O 1s peaks, shown in Figure 5c, were noticed at 532.34 eV, 532.39 eV, 531.88 eV, and 531.12 eV due to the presence of C–O–C and C=O bonds, respectively. This confirmed the existence of oxygenated functional groups present in CND, such as carbonyl and hydroxyl groups. For the 1-CND/PANI nanohybrids, the C 1s spectrum, shown in Figure 5e, revealed its peak at 284.71 eV, 284.84 eV, 284.54 eV, 285.74 eV, and 291.21 eV, which confirmed the occurrence of C=C, C–H, and C=O groups present in CND, as well as PANI, and showed a shift due to the presence of PANI. The O1s spectrum, shown in Figure 5f, showed peaks at 531.59 eV, 531.57 eV, 531.94 eV, 535.80 eV, and 536.11 eV due to the presence of C–O–C, intercalated water, and C–OH bonds, confirming the integration of oxygen-containing functional groups from the CNDs into the hybrid material. The N 1s spectrum, shown in Figure 5g, showed peaks at 401.58 eV, 401.42 eV, 400.19 eV, and 400.42 eV due to the presence of azide, imine, and NC–CH_2_–CN linkages, confirming amine functionalities within the PANI backbone. In the case of 2-CND/PANI nanohybrids, the C1s spectrum, shown in Figure 5i, showed peaks at 284.64 eV, 284.74 eV, 284.84 eV, 285.44 eV, and 287.65 eV due to the presence of C–C, C=C, and C=O bonds, respectively [27,28,29]. The O 1s spectrum, shown in Figure 5j, showed multiple peaks associated with the presence of C=O, C–OH, and C–O–C bonds of CND and PANI. The N1 s peak also showed broadening, as shown in Figure 5k, indicating the presence of higher N content and its interaction with the C of CND via N-N, N=N, and O=C–N–C=O bonds.

### 3.6. SEM with Elemental Mapping Analysis

The Fe-SEM of CND, shown in Figure 6a, shows the formation of crusty aggregates that appear to be crystalline and agglomerated. The Fe-SEM of 1-CND/PANI revealed large clusters higher in aggregation as compared to pristine CND, while the SEM of 2-CND/PANI showed flower-like deposits on large rod-like structures, which appeared to be PANI. The flaky clusters of CNDS appeared in the SEM of 1-CND/PANI and 2-CND/PANI, showing that the carbon nanodots were highly agglomerated in nature.

The FE-SEM of CND, shown in Figure 7a, shows the formation of flaky aggregates, while the elemental mapping of C, O, and N, shown in Figure 7b–d, shows that the major surface is covered with C and O and slightly with N from urea, which was used as a precursor for the synthesis of the CND. From the EDX spectrum, shown in Figure 7e, the weight percent of CND was calculated to be 74 wt.%, 12 wt.%, and 14.9 for C, O, and N, respectively, confirming the stoichiometric ratio. The Fe-SEM of 1-CND/PANI, shown in Figure 7f, exhibited the presence of elongated rods of PANI fused together with the flaky crystalline aggregates of CND. The elemental mapping, shown in Figure 7g–j revealed the surface] to be predominantly rich in C and O, as well as N and S, due to the presence of sulfur-doped PANI. The EDX spectrum, shown in Figure 7k, confirmed the C, O, N, and S loading to be 53 wt.%, 7 wt.%, 24 wt.%, and 9 wt.%, respectively. The surface was found to be homogenously composed of C, O, and N. The Fe-SEM 2-CND/PANI, shown in Figure 7l, revealed flaky particles covering the surface of PANI rods and the elemental mapping, shown in Figure 7m–p, showed the surface to be rich in C, O, and N content. The EDX spectrum, shown in Figure 7q, showed the C content to be 53%, while the N, O, and S were found to be 12 wt.%, 9 wt.%, and 15 wt.%, indicating the surface to be rich in O and S.

### 3.7. Cyclic Voltammetry and Electrochemical Sensing of CND/PANI Composites

Figure 8a,b shows the typical CV behavior with redox peaks of PANI and CND/PANI demonstrating the reversible electrochemical process. As ethanol volume increases, there is a slight variation in the current response at −0.4 V (reduction) and 0.2 V (oxidation) in PANI, suggesting that ethanol concentration influences the electrochemical activity of the material. This could be attributed to changes in the conductivity, electron transfer kinetics, or the interaction between ethanol and the electrode surface. Both graphs display similar redox behavior, but the second graph exhibits slightly higher peak currents, indicating improved conductivity or electrochemical activity when CNDs are included. PANI, as shown in Figure 8a, has a narrower range and slightly less pronounced variations in current as ethanol concentration changes. The redox peak at 0.2 V showed pronounced changes. The inclusion of CNDs likely enhances the interaction between the CND/PANI and ethanol, leading to a more pronounced electrochemical response. This comparison highlights the potential role of CNDs in enhancing performance in electrochemical systems. The oxidation peak was associated with the transition between the leucoemeraldine and emeraldine state of the polymer, that slightly shifts to higher potentials as compared to pristine PANI. The incorporation of CND in PANI withdraws electrons from the aromatic ring, making the amine units more difficult to oxidize, and can also be related to the encapsulation by CNDs, which leads to the broadening of redox peaks in the composite.

### 3.8. Electrochemical Sensing of Ethanol

The DPV plot of pristine PANI, as shown in Figure 9a, demonstrates the impact of the addition of ethanol on electrochemical behavior. The black curves, representing the system without ethanol, serve as the baseline, showcasing the inherent electrochemical behavior of the material. As ethanol is introduced in varying volumes, slight modifications in the current density are observed, particularly in the negative potential region. The more negative current density with higher ethanol volumes suggests an enhancement in surface reactivity or conductivity, potentially due to ethanol’s interaction with the polymer matrix or its influence on the electrochemical environment at the electrode interface. These changes in the current are observed to be slightly higher in 2-CND/PANI, shown in Figure 9b, due to the presence of CNDs. The sensitivity values for PANI and 2-CND/PANI samples are evaluated as 90 and 136 µA µM^−^^1^ cm^−^^2^, respectively [30,31], using the slope values from Figure 9c. The corresponding limits of detection (LOD) were found to be 0.356 and 0.325 µM, respectively. These results confirm the improved electrochemical properties of CND/PANI.

This behavior highlights the sensitivity of the PANI and CND/PANI towards ethanol, which could have implications for its use in the sensing of alcohol and the generation of alcohol during fermentation, which can help in designing smart packaging materials. The chronoamperometry profile (for an applied potential of −0.4 V) (Figure 9d) revealed the sensitivity of CND/PANI for three different alcohols (Appendix A. Cyclic voltammograms of (a) PANI, and (b) 2-CND/PANI using methanol as an analyte). The three alcohols behave differently, and they will be explored further in future.

## 4. Conclusions

This study successfully demonstrates the potential of polyaniline (PANI)/carbon nanodot (CND) composites as promising materials for smart packaging applications. The synthesis and characterization of the CND/PANI nanohybrids revealed their enhanced electro-optical and morphological properties, which were attributed to the synergistic interaction between the conducting polymer and the nanodots. Electrochemical studies further validated the effectiveness of the composites for ethanol sensing, a crucial indicator of food spoilage. The CND/PANI composites exhibited higher sensitivity, better current response, and stable redox behavior compared to pristine PANI. The incorporation of CNDs enhanced the interaction with ethanol, leading to improved sensing performance, which is pivotal for real-time monitoring of food freshness. We have observed different behavior for sensing response towards ethanol and methanol. Overall, this work establishes the feasibility of CND/PANI composites as cost-effective, environmentally friendly, and efficient materials for intelligent packaging solutions. Future studies can focus on optimizing the composite formulation and exploring its integration into commercial packaging systems for broader applications in food preservation and spoilage detection.

## Figures and Tables

**Figure 1 nanomaterials-15-00593-f001:**
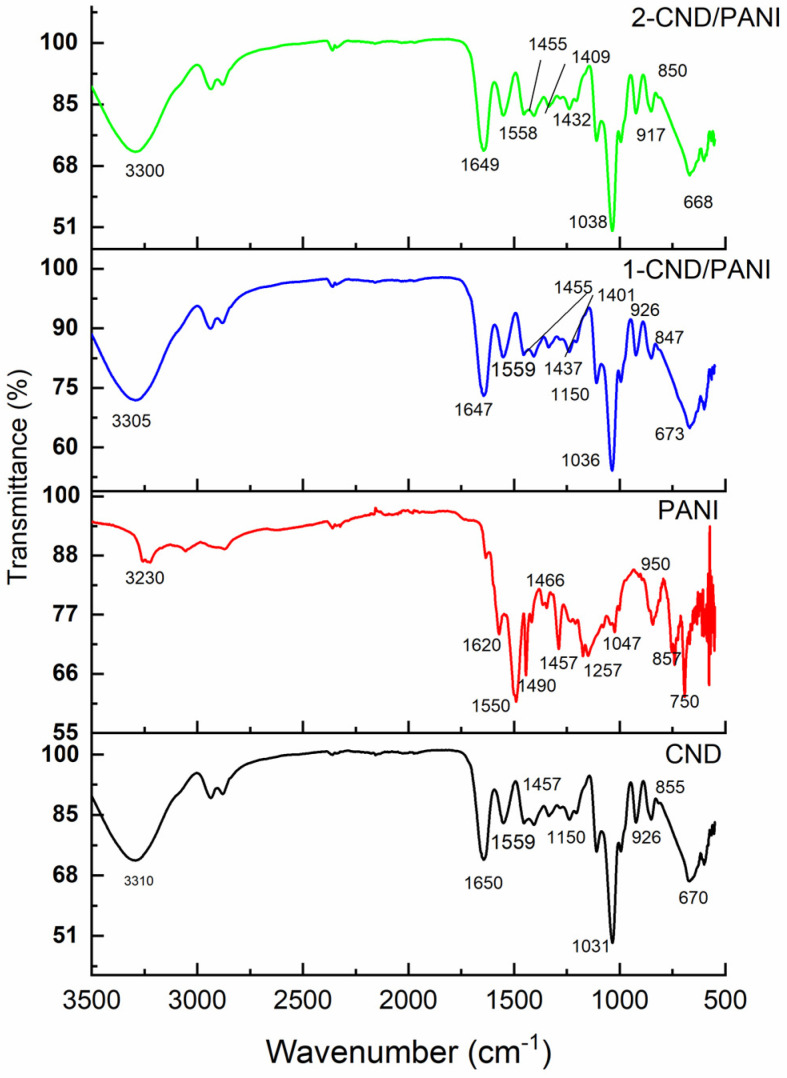
FTIR spectra of CND and CND/PANI composites.

**Figure 2 nanomaterials-15-00593-f002:**
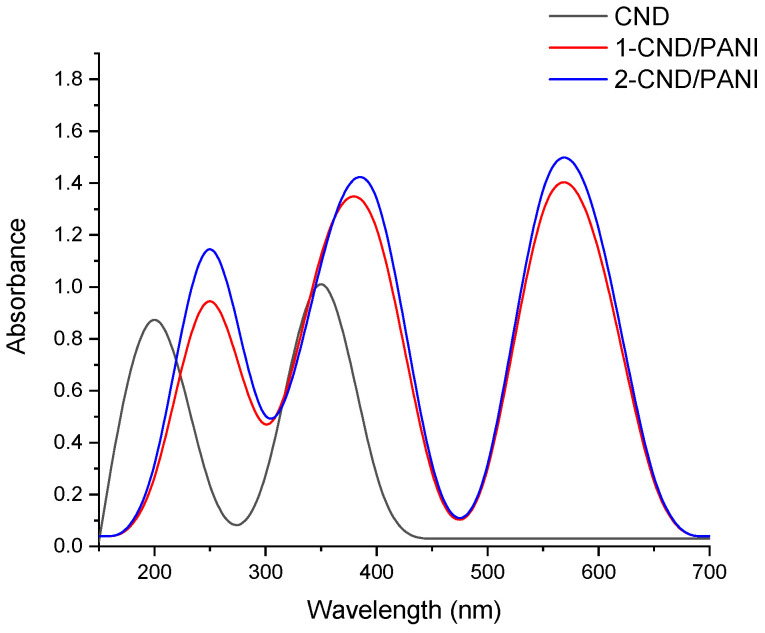
UV spectra of CND and CND/PANI nanohybrids.

**Figure 3 nanomaterials-15-00593-f003:**
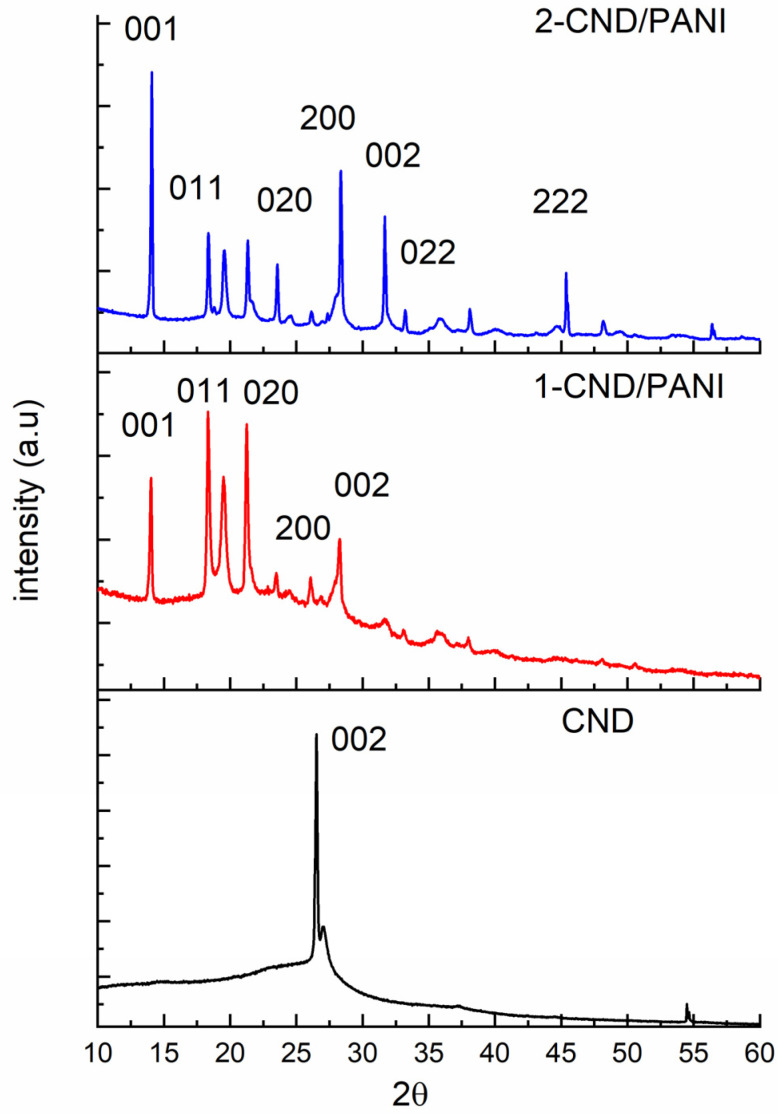
XRD profile of CND and CND/PANI composites.

**Figure 4 nanomaterials-15-00593-f004:**
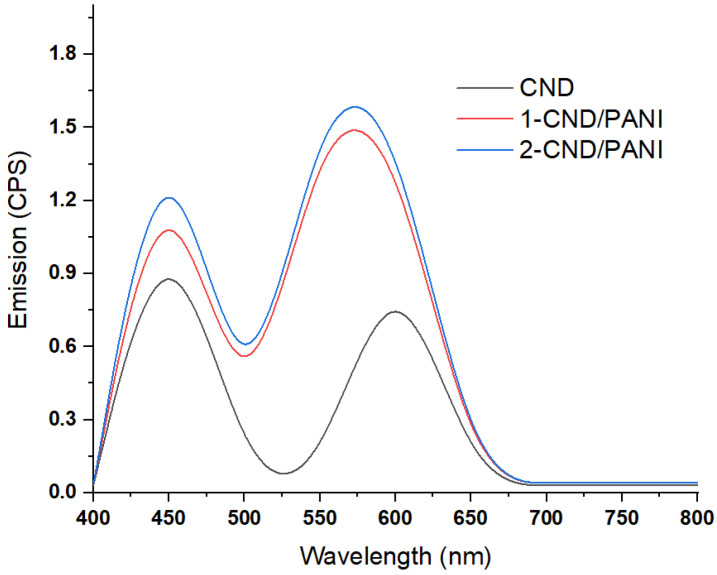
Fluorescence spectra of CND/PANI nanohybrids.

**Figure 5 nanomaterials-15-00593-f005:**
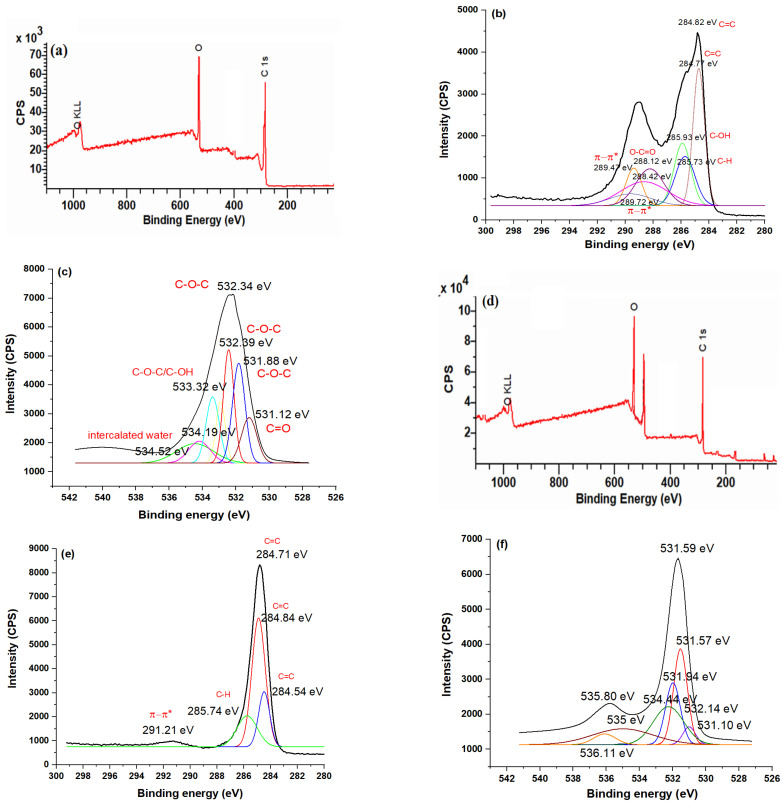
XPS survey spectrum of (**a**) CND, (**b**) C 1s CND, (**c**) O-1s CND, (**d**) XPS survey spectrum of 1-CND/PANI (**e**), C-1s of 1-CND/PANI, (**f**) O 1s of 1-CND/PANI), (**g**) N 1s of 1-CND/PANI, (**h**) XPS survey spectrum of 2-CND/PANI, (**i**) C 1s 2-CND/PANI, (**j**) O 1s 2-CND/PANI, (**k**) N 1s of 2-CND/PANI.

**Figure 6 nanomaterials-15-00593-f006:**
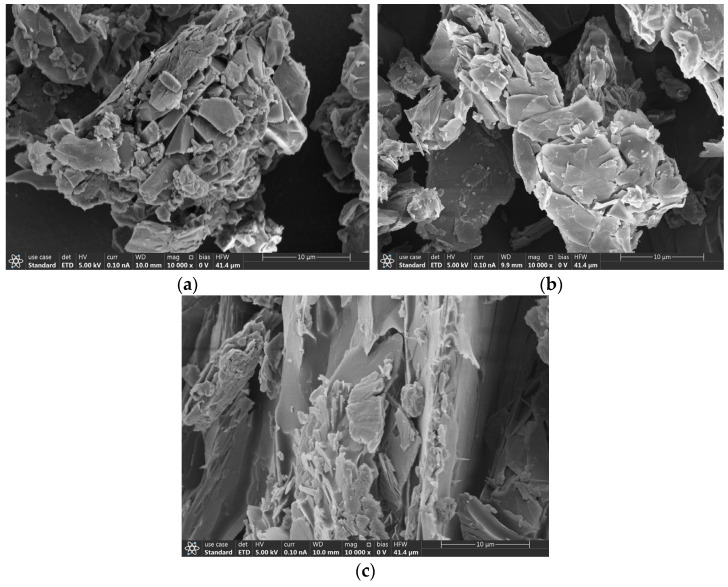
SEM of (**a**) CND, (**b**) 1-CND/PANI, (**c**) 2-CND/PANI.

**Figure 7 nanomaterials-15-00593-f007:**
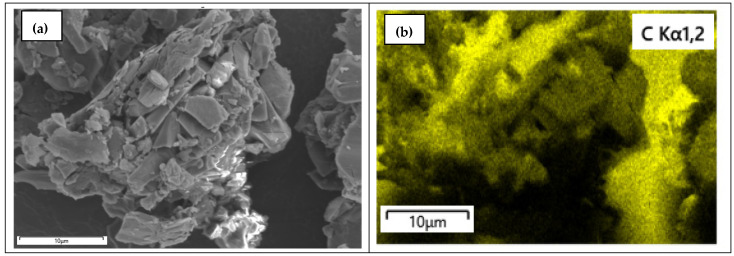
(**a**) SEM of CND, (**b**) elemental mapping of C, (**c**) elemental mapping of O, (**d**) elemental mapping of N, (**e**) EDX of CND, (**f**) SEM of 1-CND/PANI, (**g**) elemental mapping of C, (**h**) elemental mapping of O, (**i**) elemental mapping of N, (**j**) elemental mapping of S, (**k**) EDX of 1-CND/PANI, (**l**) SEM of 2-CND/PANI, (**m**) elemental mapping of C, (**n**) elemental mapping of O, (**o**) elemental mapping of N, (**p**) elemental mapping of S, (**q**) EDX of 2-CND/PANI.

**Figure 8 nanomaterials-15-00593-f008:**
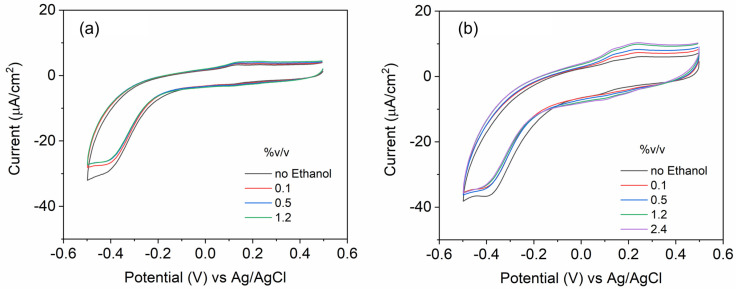
Cyclic voltammograms of (**a**) PANI, and (**b**) 2-CND/PANI.

**Figure 9 nanomaterials-15-00593-f009:**
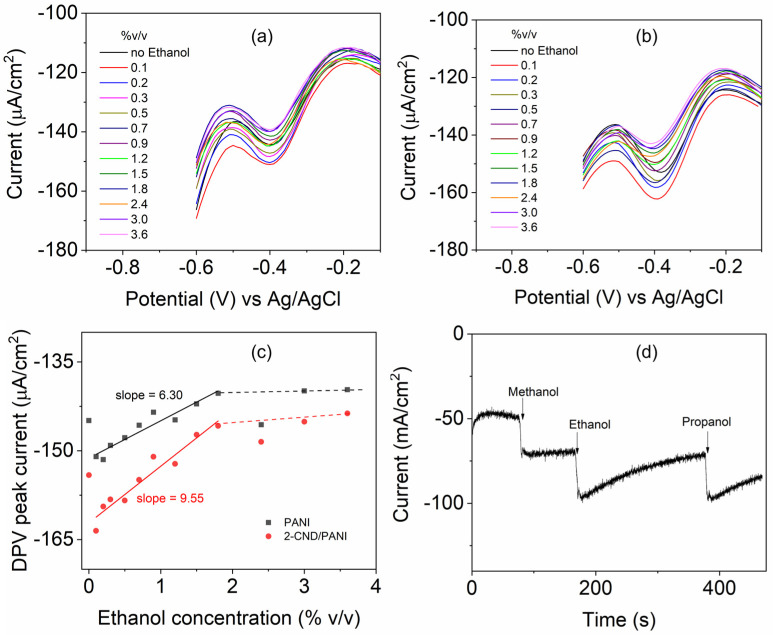
Current–voltage plots of (**a**) pure PANI and (**b**) 2-CND/PANI composite. (**c**) Calibration plot of DPV peak current with respect to ethanol concentration. (**d**) Chronoamperometric detection of different alcohols by CND/PANI.

## Data Availability

Data are reported in this manuscript.

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
