# Peer review of "Synthesis and Characterization of Polyaniline/Carbon Nanodots: Electrochemical Sensing of Alcohols for Freshness Monitoring for Application as Packaging Materials"

_nanomaterials, 2025, doi:10.3390/nano15080593_

Round 1

Reviewer 1 Report

Comments and Suggestions for Authors

Jackson et al. described synthesis and characterization of polyaniline/carbon nanodots: for electrochemical sensing of alcohols.  Manuscript is slightly improved than its previous version but still main issue is same which is sensor sensing performance. My comments are below.

  1. Introduction: It is not mentioned that which alcohol is produced when food get spoiled. Moreover, introduction lacks information about how food spoilage in general or with sensors are measured. If no technology exists, it should be clearly mentioned.
  2. It is not mentioned that in what condition DPV was recoded, after argon/nitrogen purging? Please this information in experimental section.
  3. Why not morphological characterization by TEM is added of CND? What was the size of those carbon dots?
  4. Fig. 1; all spectra appear exactly same. Please also add PANI IR spectra to find specific peaks for PANI.
  5. Fig. 2; Add here also PANI alone. Origin of all peaks should be added.
  6. Fig. 3; XRD: Why CND 002 peak was shifting?
  7. Fig. 9; Current-voltage plots are still confusing. Was it decreasing after addition of ethanol? Author mentioned in text that "The increased current density with higher ethanol volumes suggests an enhancement in surface reactivity or conductivity, potentially due to ethanol's interaction with the polymer matrix or its influence on the electrochemical environment at the electrode interface" Curve suggest decrease after ethanol addition. Please clarify this inconsistency.
  8. Selectivity, chronoamperometric curve suggest that binding or interaction of ethanol is irreversible that is why current change was not stable. However, for methanol it was stable. As I can see sensor shows better response for methanol. Did author tried to measure methanol different concentration and prepared calibration curve?
  9. This text is not clear, the chronoamperometry profile Figure 9 (c) revealed the sensitivity of CND/PANI detected at different voltages confirming that the detection was selective as it was noticed at 3 different current values” Where are those different potential results? Fig. (c describe the electrochemical analysis performed at PANI and PANI-CND electrodes.
  10. Conclusion lines 331-332; it seems incorrect in view of result provided in Fig. 9D.
  11. Supporting information file is missing.
  12. ref 30, Author name is wrong.

Author Response

  1. Introduction: It is not mentioned that which alcohol is produced when food get spoiled. Moreover, introduction lacks information about how food spoilage in general or with sensors are measured. If no technology exists, it should be clearly mentioned.

A: The text has been modified.

  1. It is not mentioned that in what condition DPV was recoded, after argon/nitrogen purging? Please this information in experimental section.

A: The text has been modified.

  1. Why not morphological characterization by TEM is added of CND? What was the size of those carbon dots?

A: The CNDs were ranging between 130 nm to 150 nm which was calculated by the SEM micrographs. TEM could not be done due to lack of availability of the instrument under working condition.  

  1. Fig. 1; all spectra appear exactly same. Please also add PANI IR spectra to find specific peaks for PANI.

A: PANI spectrum has been added. Peaks appear same due to low loading of CNDs.

  1. Fig. 2; Add here also PANI alone. Origin of all peaks should be added.

A: This has been added and discussed.

  1. Fig. 3; XRD: Why CND 002 peak was shifting?

A: The shifting was attributed to reorganization of planes due to the presence of PANI.

  1. Fig. 9; Current-voltage plots are still confusing. Was it decreasing after addition of ethanol? Author mentioned in text that "The increased current density with higher ethanol volumes suggests an enhancement in surface reactivity or conductivity, potentially due to ethanol's interaction with the polymer matrix or its influence on the electrochemical environment at the electrode interface" Curve suggest decrease after ethanol addition. Please clarify this inconsistency.
  • The DPV response is recorded without cathodic inversion. In this mode, the peaks appear in the negative region, instead of conventional upward peaks. The ‘increased current density’ meant ‘more negative’ and has been replaced in the revised text to avoid confusion. It shows enhancement in current value with addition of ethanol.
  1. Selectivity, chronoamperometric curve suggest that binding or interaction of ethanol is irreversible that is why current change was not stable. However, for methanol it was stable. As I can see sensor shows better response for methanol. Did author tried to measure methanol different concentration and prepared calibration curve?
  • As suggested, we have now carried out the CV and DPV analyses using methanol. It is included in the supplementary information file. However, the composite material doesn’t show good DPV response, compared to pristine.
  1. This text is not clear, the chronoamperometry profile Figure 9 (c) revealed the sensitivity of CND/PANI detected at different voltages confirming that the detection was selective as it was noticed at 3 different current values” Where are those different potential results? Fig. (c describe the electrochemical analysis performed at PANI and PANI-CND electrodes.

A: We are thankful to the reviewer for pointing this out. We have corrected the text. The chronoamperometry was carried out at one fixed potential (-0.4 V) using three different alcohols (150 µL each).

  1. Conclusion lines 331-332; it seems incorrect in view of result provided in Fig. 9D.

A: We have rectified the same.

  1. Supporting information file is missing.

A: We have now added the supporting information file.

  1. ref 30, Author name is wrong.

A: The ref. is corrected.

Reviewer 2 Report

Comments and Suggestions for Authors

The manuscript can be accepted in the revised form.

Author Response

We thank the reviewer for accepting the changes and revision. 

Round 2

Reviewer 1 Report

Comments and Suggestions for Authors

I am satisfied with Author's response to my comments.